# LEARNING WEIGHT SENSITIVITY FROM ENTROPY

## ABSTRACT

Multiple network pruning methods have used connection sensitivity of each weight to prune their network. This paper proposes a meta-learning approach to learn sensitivity of weights based on their entropy, or change, during the training phase. We have experimentally shown the validity of such an approach.

## 1 INTRODUCTION

Convolutional Neural Networks (CNNs) have been shown to achieve state-of-the-art results in various computer vision tasks. However, all CNNs currently in use are padded with filters and weights that have no impact on the final predictions. There have been no methods to identify these weights and filters prior to creation of networks which would improve efficiency of the generated network.

Lottery Ticket Hypothesis (Frankle & Carbin, 2018) states that a trained dense neural network contains a sub-network, which when trained in isolation, can produce results equal, if not superior, to the original trained network. Various algorithms have been proposed, which have pruned CNN models down significantly (Xu et al., 2020), which shows that CNN models are larger than they need to be.

Our contribution lies in three folds:

- We proposed a method to train networks to identify patterns that emerge during training to identify criterion based on connection sensitivity.
- We experimentally show that the position of a weight in a network is as important in determining the effectiveness as the value it holds.

## 2 PROPOSED METHOD

Training a network involves tuning each individual weights of a network to function from its initial random values to new weight that give a desired output. This research intends to prove that the position of a weight is more important than the initial value. The experiment was conducted in two phases; Computing the Entropy, and recreating a network after learning from the Entropy.

### 2.1 COMPUTING ENTROPY

The Entropy represents the change in magnitude of weights inside the filter. For each filter, individual and separate Entropy arrays are created after each training epoch. The separate filters are then used to create the new network using meta-learning. The algorithm for learning sensitivity from each filter of the trained network is given by the following formula:

$$\theta_{i+1}^* = \Delta\theta_{i+1} + \alpha \frac{\partial f(\theta_i)}{\partial \theta_i^*} \tag{1}$$

where $\theta$ and $\theta^*$ represent the network and the connection sensitivity of weights of the network after meta-learning from each Entropy, $\Delta\theta_i$ represents entropy of network $\theta$ after training for $i$ epochs, $f$ represents the training function, and $\alpha$ is the learning rate. After each training epoch, the loss of the network w.r.t its connection sensitivity is computed, then added to the entropy of the current network to compute the subsequent connection sensitivity. As can be seen in Equation1, the final network is meta-learned from entropy of the network in every training.

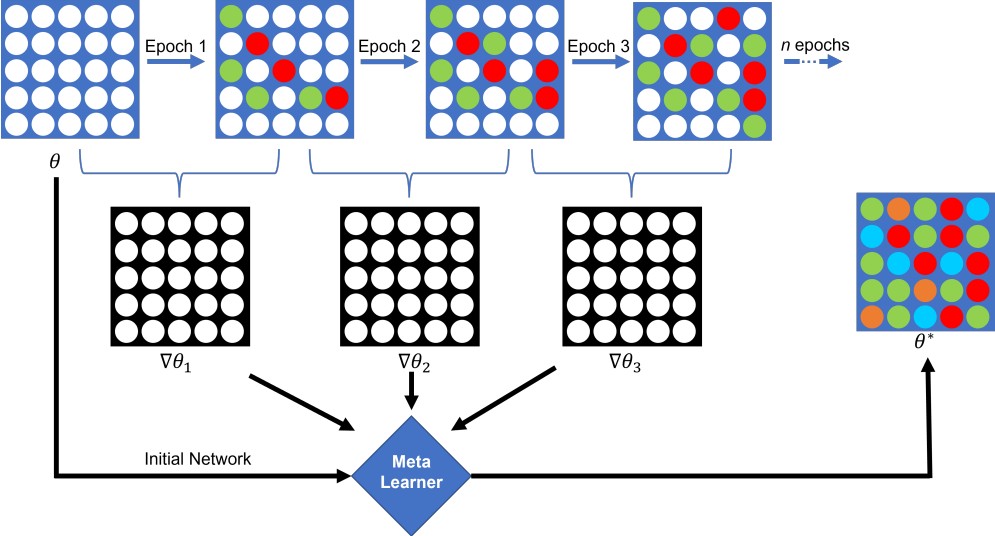

Figure 1: Algorithm of the method to learn connection sensitivity of each weight.

Table 1: Comparison of Entropy in various layers after 160 epochs on two different VGG-16 (Simonyan & Zisserman, 2014) models for ImageNet dataset (Deng et al., 2009). Learned Entropy refers to the slight changes to the weight value, and Absolute Entropy is the large changes.

| Layer Number | Different Weights | | Same Weights | |
|:---:|:---:|:---:|:---:|:---:|
| | *Learned* | *Absolute* | *Learned* | *Absolute* |
| 0 | 0.00% | 0.06% | 0.00% | 0.00% |
| 1 | 5.68% | 50.10% | 16.09% | 28.22% |
| 2 | 0.03% | 0.04% | 0.01% | 0.08% |
| 7 | 0.02% | 0.18% | 0.04% | 0.00% |
| 12 | 12.41% | 50.05% | 8.88% | 9.09% |

## 2.2 PRUNING AND RETRAINING

Once the entropy has been learned, the final network gives the sensitivity of each weight in each filter of the trained network. A new network can be created by pruning the untrained original network based on the weight sensitivities as learned from entropy of the same trained network. This significantly reduces the number of FLOPs necessary to train and use the network.

## 3 EXPERIMENTS

We trained multiple models with the same dataset, but the batches contain random images, all of which are also randomly augmented. Some models have the same initial values, while some start with different values. As can be seen from Table 3, the difference between weights learned when after training is not very different when starting with the same weights. This proves that when using pre-trained weights, the position of filters and weights are more important than their values. Hence, it would be more advantageous to find ideal network configurations than to find ideal initial weights.

## 4 CONCLUSION

In this work, we have presented the following: 1) We used entropy of the networks during training to identify patterns among weights. 2) We used used these patterns to compute the connection sensitivity of weights based on their position in filters. 3) We experimentally show that the position of weights in a network is a factor in determining its sensitivity.

## 5 URM STATEMENT

The author acknowledges that this work meets every URM criteria of ICLR 2023 Tiny Papers Track. This is also their first paper ever to be submitted, and has been done without any notable research funding.

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
