# OpenReview forum: "Learning Weight Sensitivity from Entropy"
_ICLR.cc/2023/TinyPapers — Submitted to Tiny Papers @ ICLR 2023_

### Official Review · Reviewer_QtPb · 2023-03-28

**Confidence:** 4

**Summary Of Contributions:**

This paper tries to analyze the entropy of weights in CNNs. Unfortunately, due to unclear and confusing writing, I am not able to understand the paper in more details.

**Rating:**

Needs Clarification (NC): a submission which does not meet the reviewing criteria and needs clarification for its described problem or solution

**Strengths And Weaknesses:**

Strengths: Unclear.

Weaknesses:
- Clarity: Unfortunately, the writing is very unclear and confusing. It is unclear how this paper computes the entropy, what "connection sensitivity" is, how pruning is conducted, how the models in the experiments are configured, what the identified patterns are, how "position of weights" is reflected in the method and experiments, etc. The conclusions are also unclear.
- Many arguments are too strong without sufficient support. E.g., "it would be more advantageous to find ideal network configurations than to find ideal initial weights"; "However, all CNNs currently in use are padded with filters and weights that have no impact on the final predictions.".


**Suggested Changes:**

I would suggest the authors to significantly improve the clarity of the writing, and make the arguments weaker if there is no sufficient evidence.

---

### Official Review · Reviewer_c3C1 · 2023-03-30

**Confidence:** 2

**Summary Of Contributions:**

This paper proposes a meta-learning approach to learn the sensitivity of weights in a neural network based on their entropy during training. The authors experimentally show that the position of weights in a network are more important their values.

**Rating:**

Clear, Correct, and Reproducible (CCR): a submission which meets the reviewing criteria

**Strengths And Weaknesses:**

# Strengths
1. The method proposed in this paper can be used to organize and optimize neural networks more efficiently, thereby improving the performance of computer vision and other related tasks.
2. The method is a novel meta-learning approach that identifies sensitive locations of weights and filters to more efficiently prune and retrain networks.
3. The paper shows experimentally that with pretrained weights, filters and weight positions are more valuable than their actual values. Therefore, finding the ideal network configuration is more advantageous than finding the ideal initial weights.

# Weaknesses
1. The paper has some mistakes, such as 'Table 3` in the *Section 3. EXPERIMENTS*, but there is only one table in the paper.
2. The detailed description of the experiment and method in the paper is too simple, which is easy to cause some confusion.
3. The paper does not give the experimental results of the effect and efficiency comparison between the method before pruning and the method after pruning.

**Suggested Changes:**

The author should revise and optimize the paper according to the content mentioned in the *Weaknesses*.

---

### Official Review · Reviewer_oqmM · 2023-03-31

**Confidence:** 4

**Summary Of Contributions:**

This paper proposes a meta-learning approach to learn sensitivity of weights based on their entropy, or change, during training.

**Rating:**

High Potential (HP): a submission which meets the reviewing criteria and has potential to make an impact on the field

**Strengths And Weaknesses:**

Strengths:
i) The authors use entropy of the networks during training to identify patterns among weights.
ii) The authors use these patterns to compute the connection sensitivity of weights based on their position in filters.
iii) Results show that the position of weights in a network is a factor in determining its sensitivity.

Weaknesses:
i) Contributions in Introduction
ii) Some typos.


**Suggested Changes:**

i) In Introduction, the authors mention that the contribution lies in three folds. However, the authors list two contributions. It is recommended that the authors extend the first contribution into two ones (similar to conclusion).
ii) Please proofread the text. For instance, in Conclusion it is mentioned that: "used used"

---

### Meta-Review · Area_Chair_NLER · 2023-04-08

**Recommendation:** Invite to revise
**Confidence:** 3

**Metareview:**

This paper presents an approach for pruning a neural network using metalearning, by looking at their "entropy" (how they change during training) to estimate the sensitivity of the network to the weights.

Using the change in magnitude of weights to identify sensitive weights seems like an interesting way to prune neural networks. Reviewers oqmM and c3C1 note that this could be used to more efficiently train networks and could have impact on the field, and that the dependence of sensitivity on weight position is an interesting finding as well. On the other hand, all of the reviewers have noted some concerns with the clarity of the proposed approach, and I also found many of the details to be hard to understand.

Overall, this paper seems like a good start and is tackling an interesting problem, but I think it should be revised to ensure that it is clear and reproducible.

**Summary:**

The paper proposes a promising approach for learning sensitivity of weights and applying this to pruning. Reviewers note some advantages of this approach, but also a number of issues with clarity and reproducibility.

**Reason For Not Giving A Higher Recommendation:**

Overall, the reviewers have identified a number of clarity and reproducibility concerns that should be addressed before this paper can meet the Clarity, Correctness, and Reproducibility (CCR) criteria.

The reviewers noted these concerns:

- The contribution statement is somewhat unclear (it says there are three contributions, but only two are listed).
- The description of the method is somewhat confusing, and some of the terms are not defined.
- The experiments are not described in much detail, which could make it difficult to reproduce the findings. It might also be worthwhile to perform comparisons between pruned and unpruned networks, in addition to just looking at entropy.
- Some of the claims about prior work are not well supported with evidence and citations.
- Some typos.

To make this work clearer and more reproducible, I think it would be worth expanding section 2.1 to give more information on:

- what the authors mean by "connection sensitivity",
- how the values $\Delta \theta_{i+1}$ and $\frac{\partial f(\theta_i)}{\partial \theta_i^*}$ are computed,
- how the sensitivity and entropy are used to construct a final pruned network.

and revising section 3 to clarify how the experiments were configured and how the results in Table 1 support the conclusions drawn.

**Reason For Not Giving A Lower Recommendation:**

N/A

---

### Decision · Program_Chairs · 2023-04-09

No revision received; not invited to archive